# Experimental and Theoretical Studies on Extract of Date Palm Seed as a Green Anti-Corrosion Agent in Hydrochloric Acid Solution

**DOI:** 10.3390/molecules26123535

**Published:** 2021-06-09

**Authors:** Naba Jasim Mohammed, Norinsan Kamil Othman, Mohamad Fariz Mohamad Taib, Mohd Hazrie Samat, Solhan Yahya

**Affiliations:** 1Materials Science Program, Department of Applied Physics, Faculty of Science and Technology, University Kebangsaan Malaysia, Bangi 43600, Selangor, Malaysia; Naba_jasem@yahoo.com; 2Faculty of Applied Sciences, Universiti Teknologi MARA (UiTM), Shah Alam 40450, Selangor, Malaysia; 3Ionic Materials & Devices (iMADE) Research Laboratory, Institute of Science, Universiti Teknologi MARA (UiTM), Shah Alam 40450, Selangor, Malaysia; mohdhazrie@uitm.edu.my; 4Faculty of Applied Sciences, University Technology MARA (UiTM) Perlis Branch, Arau 02600, Perlis, Malaysia; solhan@uitm.edu.my

**Keywords:** date palm seed (DPS) extract, green anti-corrosive, weight loss test, density functional theory (DFT), HCl

## Abstract

Extracts from plant materials have great potential as alternatives to inorganic corrosion inhibitors, which typically have harmful consequences. Experimental and theoretical methodologies studied the effectiveness of agricultural waste, namely, date palm seed extract as a green anti-corrosive agent in 0.5 M hydrochloric acid. Experimental results showed that immersion time and temperature are closely related to the effectivity of date palm seed as a corrosion inhibitor. The inhibition efficiency reduced from 95% to 91% at 1400 ppm when the immersion time was increased from 72 h to 168 h. The experimental results also indicated that the inhibition efficiency decreased as the temperature increased. The presence of a protective layer of organic matter was corroborated by scanning electron microscopy. The adsorption studies indicated that date palm seed obeyed Langmuir adsorption isotherm on the carbon steel surface, and Gibbs free energy values were in the range of −33.45 to −38.41 kJ·mol^−1^. These results suggested that the date palm seed molecules interacted with the carbon steel surface through mixture adsorption. Theoretical calculations using density functional theory showed that the capability to donate and accept electrons between the alloy surface and the date palm seed inhibitor molecules is critical for adsorption effectiveness. The HOMO and LUMO result indicated that the carboxyl (COOH) group and C=C bond were the most active sites for the electron donation-acceptance type of interaction and most auxiliary to the adsorption process over the Fe surface.

## 1. Introduction

Recently, carbon steel corrosion study has attracted the attention of researchers in the corrosion field. Since carbon steel is applied widely in oil and gas applications, it is usually placed under the sea, which contains corrosive agents, including water, acids, salts, humidity, and oxygen. These agents increase the corrosion reactions on carbon steel surfaces [1]. Many methods have been proposed to overcome these corrosion problems, such as coating, cathodic protection, anodic protection, and corrosion inhibitors. Corrosion inhibition is one of the common effective ways to overcome corrosion reactions, especially in acidic media. The corrosion rate is reduced when molecules are adsorbed onto the metal surface, thereby decreasing anodic or cathodic reactions [2].

Commonly used corrosion inhibitors are synthetic chemicals, including chromates, lead oxide, zinc phosphates, polyphosphates and orthophosphates. These inhibitors, produced synthetically, do not decompose easily and require costly, time-consuming, and energy-intensive processes to dissolve [3]. As a result of the environmental and safety issues associated with inhibition processes, researchers have begun to look at natural inhibitors that are less harmful to the environment [4].

Many researchers are working to develop environmentally friendly and natural inhibitors known as green corrosion inhibitors. Green inhibitors are non-toxic, natural, friendly to the environment, and renewable. These compounds generally contain nitrogen, sulphur, and oxygen and can be extracted inexpensively from plant parts such as leaves, seeds, flowers, and fruits using simple methods [5,6]. The effectivity of these inhibitors depends on their electronic structures, electron density at the donor sites, aromaticity, molecular area, type of corrosive media, and temperature.

The topic of green corrosion inhibitors has been widely published in many works of literature. The extract of *Saraca ashoka* was used as a corrosion inhibitor for mild steel and found to give 89.9% protection in impedance measurements [7]. *Papaver somniferum* was employed to inhibit stainless steel with an 83% of inhibition efficiency [8], whereas *Ginkgo* leaf extract was utilised to inhibit X70 steel, and the IE reached up to 89% in an impedance study at normal temperature [9]. An extract of *Polygonatum odoratum* was applied to inhibit aluminium corrosion with a maximum protection of 72% [10]. Date palm fruit juice protected aluminium with a top IE of 72% at a concentration of 2000 ppm [11]. Date palm leaf extract afforded protection up to 80% against HCl-induced corrosion [12]. Simultaneously, date palm fibre was used to protect carbon steel, and the maximum IE was 84% [13]. Meanwhile, date palm seed extracts are capable of inhibiting the corrosion of carbon steel with 59% of efficiency [14]. At a concentration of 1.5 gL^−1^, date palm extract was capable of inhibiting the corrosion of stainless steel with an efficiency of 88% [15].

In our previous study, an extract of date palm seed gave an inhibition of carbon steel corrosion ranging from 94% to 97% using electrochemical impedance spectroscopy and potentiodynamic polarization measurments, respectively [16]. Fatty acids are attractive candidates as corrosion inhibitors because they contain absorbable carboxyl functional groups for attachment to metal surfaces. Besides, they are readily available and more environmentally friendly. Gas chromatography-mass spectrometry (GC-MS) results reportted by our research group show that DPS contains high levels of different acids, which formulas are indicated in Table 1 [16]. The importance of this class of acids is due to their molecular features. Conceptually, the carboxylic acid functional group in fatty acids enhances the surface chemistry toward interaction with metal-vacant orbitals and plays a crucial role in improving the adsorption characteristics. The active group of fatty acid molecules may interact with the metal protecting it from corrosion.

There have been no previous reports on the use of DPS as a green anti-corrosion agent on carbon steel in HCl solution using theoretical studies. Theoretical studies are a powerful tool for corrosion inhibition research to measure and analyse the electronic properties of the molecules involved and examine the reaction mechanism(s) involved [17]. Studies describing the electronic properties of corrosion inhibitors are limited. Computational techniques can effectively propose the best compound in corrosion inhibitors and overcome any experimental disadvantages. Therefore, the application of computational methods is an economical, fast, and eco-friendly way to complement experimental investigations.

The main purpose of this article was to evaluate DPS as a carbon steel corrosion inhibitor in 0.5 M HCl solution, using weight loss at various temperatures and immersion time to measure the protective effect. The kinetics and thermodynamics parameters were calculated and discussed. Scanning electron microscopy (SEM) was used to analyse the inhibited and uninhibited carbon steel surface in HCl solutions. DFT methods were used to study the relationship between the inhibitory properties of DPS and the structure of its components. DFT methods were also used to compute the Fukui indices, Highest Occupied Molecular Orbital (E_HOMO_) and Lowest Unoccupied Molecular Orbital (E_LUMO_) energies.

## 2. Materials and Methodologies

### 2.1. Carbon Steel Preparation

The chemical composition of the tested carbon steel (wt. %) was 99.3 Fe; 0.5 Mn; 0.04 P; 0.12 C; 0.045 S. The carbon steel was mechanically cut into pieces measuring 2 × 2 cm. The samples were polished using silicon-carbide paper with #320–#2000 grades. The samples were finally washed with acetone and then distilled water before use [18].

### 2.2. Extraction Process

The date seeds were washed with water and sun-dried for a few days before further drying in an oven at 40–50 °C for 20–30 min. The dried date seeds were ground mechanically into powder. In a typical DPS extraction, 10 g of dried date seed powder was diluted with 1 L of 0.5 M HCl solution and heated to 50 °C using a hot plate under continuous magnetic stirring for 15 h [19]. The solution was then filtered to eliminate any suspended solid particles. Finally, the extract was diluted with 0.5 M HCl to prepare a series of inhibitor stock solutions with different DPS concentrations (800, 900, 1200, 1400, and 2000 ppm).

### 2.3. Weight Loss Test

Weight loss was measured, firstly, at different immersion times (72–168 h) at room temperature and secondly, at various other temperatures (25, 35, 45, 55 and 65 °C) to determine the kinetic parameters of the corrosion process and elucidate the inhibition mechanism. The carbon steel samples were prepared as explained in Section 2.1. The samples were weighed before and after immersion in a corrosive medium. The data gathered from the weight loss test was applied to determine the corrosion rate ($C_{r}$) in millimetres penetration per year (mm/y^−1^) and inhibition efficiency (IE%) utilised by the equations below [20]:(1)$$C_{r}\left(mmy^{-1}\right)=\frac{87600\times \Delta W}{\rho TA}$$
where ∆W is the difference between initial and final weights of carbon steel (mg), while ρ is the density of carbon steel (g cm^−3^), parameter A is the surface area of carbon steel (cm^2^). The immersion time per hour (h) denoted as T, and 87,600 is a constant for C_r_ determination [21].
(2)$$IE\%=\left(\frac{C_{r_{blank}}-C_{r_{inh}}}{C_{r_{blank}}}\right)\times 100$$
where $C_{r_{\mathrm{blank}}}$ and $C_{r_{\mathrm{inh}}}$ are the corrosion rates values without and with DPS, respectively.

### 2.4. Scanning Electron Microscopy (SEM)

Carbon steel samples 25 × 25 cm size were exposed to 0.5 M HCl solutions in the absence and presence of 1400 ppm DPS for 24 h at 25 °C. SEM and EDX images were recorded using a MERLIN model scanning electron microscope (SEM; ZEISS, Oberkochen, Germany) located at the Universiti Kebangsaan Malaysia (UKM) CRIM lab. The SEM experiments was conducted at a voltage of 15 kV, 500× magnification and the size of each image was 10 µm. The EDX detector was used to examine the chemical compounds on the carbon steel surface.

### 2.5. Quantum Chemical Calculation

The geometrical optimisation of the inhibitor molecules was performed using the DMol3 software with the generalised gradient approximation of the Perdew-Burke-Ernzerhof (GGA-PBE) functional within the DFT framework. The double numerical with polarisation (DNP) function was used as a basis set. In all geometric optimisations, the convergence energy tolerance of 1.0 × 10^−5^ Ha, the maximum force of 0.002 Ha/Å, and the maximum displacement of 0.005 Å were used. Each of the following quantum parameters, such as energy gap (ΔE) eV, electron affinity (EA) eV, ionisation potential (IP) eV, electronegativity (χ) eV, global hardness (η) eV, global softness (σ), and (ΔN) the fraction of electrons transferred from inhibitor to Fe (110) surface was determined using the following Equations [22]:(3)$$\Delta E=E_{\mathrm{LUMO}}-E_{\mathrm{HOMO}}$$
(4)$$\mathrm{EA}=-E_{\mathrm{LUMO}}$$
(5)$$\mathrm{IP}=-E_{\mathrm{HOMO}}$$
(6)χ=(IP+EA)/2
(7)η=(IP−EA)/2
(8)σ=1/η
(9)$$\Delta N=\left(\phi -\chi_{\mathrm{inh}}\right)/\left[2\left(\eta_{\mathrm{inh}}\right)\right]$$
where ϕ equal 4.82 eV, which represents the work function of the Fe (110) surface [23].

## 3. Major Findings

### 3.1. Weight Loss Test

Weight-loss tests are straightforward since they don’t need specialised equipment and can be performed on several samples simultaneously. This approach aimed to investigate the impact of immersion time and temperature on Cr and IE%. and the details are as follows:

#### 3.1.1. The Influence of Immersion Time

The immersion periods were varied from 72 h to 168 h in this study, and the resulting influence is shown in Figure 1. It is clear that the IE% was reduced after the longest period of immersion. The data obtained for the weight loss of carbon steel in 0.5 MHCl in the absence and presence of DPS inhibitor concentrations are displayed in Table 2. A review investigation of these data shows that the weight loss increases and IE% decreases as immersion time increases. A maximum value of IE reached 95% and reduced to 91% after immersion for 72 h and 168 h, respectively. This can be attributed to DPS being desorbed from the carbon steel surface and increasing contact between the bare metal and corrosive media would lead to the metal dissolving. The decrease in IE% after a long period of immersion can also be due to a lower number of inhibitor molecules available in HCl solution to avoid or minimise metal dissolution. After the inhibitor molecules had been desorbed from the carbon steel surface, they become inactive and do not participate in the inhibition process [24,25]. These molecules will not be toxic to the environment after disposal due to their biodegradability.

#### 3.1.2. The Influence of Temperature

In this work, the effect of temperature on the inhibition of carbon steel has been studied in the range of 25 °C to 65 °C. The results illustrated in Figure 2 show the values of corrosion rate (Cr) and inhibition efficiency (IE%) obtained by the weight loss method. It can be seen from Table 3 that an increase in temperature will increase the Cr and decrease IE%. Meanwhile, with the participation of inhibitor, the Cr was reduced, and IE% increased as the concentration of inhibitor increases. At higher temperatures, the time difference between the adsorption and desorption of inhibitor molecules on the carbon steel surface becomes shorter [26]. Corrosion in an acidic medium is commonly accompanied by the release of H_2_. The adsorption process is affected by the irritation resulting from accelerated development rates of H_2_ at a high temperature, which drives a reduction in the IE% of corrosion inhibitor [27].

### 3.2. Scanning Electron Microscopy (SEM)

Figure 3 depicts the SEM images under 500× magnification of the carbon steel surface: (a) just after grinding with silicon paper, (b) in 0.5 M HCl, and (c) in 0.5 M HCl solution in the presence of 1400 ppm of DPS corrosion inhibitor. The morphology of the polished sample in Figure 3a shows a smooth surface with no visible signs of corrosion. After retrieving the sample from the uninhibited 0.5 M HCl medium, it was found to have deteriorated severely, as seen in Figure 3b. When the immersion test was repeated with the acid solution containing 1400 ppm of DPS corrosion inhibitor, the surface was observed to be relatively smooth, indicating significant corrosion suppression, as presented in Figure 3c. These outcomes suggested that the presence of DPS inhibited the corrosion of carbon steel. EDX was conducted to examine the elemental composition of the samples in the absence and presence of 1400 ppm of DPS corrosion inhibitor at 25 °C. The results of the EDX analyses are summarised in Figure 4 and show that the presence of DPS suppressed the corrosive effects on carbon steel surfaces immersed in HCl medium to a greater extent than if the carbon steel surfaces were uninhibited [28].

### 3.3. Thermodynamic and Kinetic Model

#### 3.3.1. Thermodynamics of Corrosion in HCl Solution

Temperature plays an essential role in understanding the inhibitive mechanism of the corrosion process and study the impact on corrosion rate. Therefore, the activation energy ($E_{a}$), activation enthalpy $(\Delta H_{a}$), and activation entropy ($\Delta S_{a})$ were determined at different temperature and different concentrations of DPS. The relationship between the Cr and temperature (T) follows the Arrhenius equation:(10)$$\ln\left(Cr\right)=LnA-\frac{Ea}{RT}$$
where A is a pre-exponential factor, Ea is the smallest energy for a reaction that needed to occur (activation energy), R represents the universal gas constant (8.314 J mol^−1^ K^−1^), and T represents the absolute temperature in Kelvin (K).

The values of activation energy were calculated by plotting LnCr against 1/T, as shown in Figure 5. This plot provides straight lines wherever the intercept is lnA, and the slope is −Ea/R; the values obtained are listed in Table 4. The $E_{a}$ values of DPS corrosion inhibitor were in the range of 23.62 to 83.43 kJ·mol^−1^. Therefore, it is suggested that DPS corrosion inhibitor can perform dual adsorption reactions, namely, both chemical and physical adsorption [29]. In corrosion inhibition studies, the $E_{a}$ values of 80 kJ·mol^−1^ and above indicate a chemical adsorption reaction process. On the contrary, $E_{a}$ values of less than 80 kJ·mol^−1^ represent physical adsorption [30].

Meanwhile, the values of $\Delta H_{a}$, and $\Delta S_{a}$ were determined by using the transition state, which is an alternative formula for the Arrhenius equation:(11)$$Cr=\frac{RT}{Nh}exp\left(\frac{\Delta S_{a}}{R}\right)exp\left(-\frac{\Delta H_{a}}{RT}\right)$$
(12)$$ln\left(\frac{Cr}{T}\right)=-\frac{\Delta H_{a}}{RT}+\left[ln\left(\frac{R}{Nh}+\frac{\Delta S_{a}}{R}\right)\right]$$

N is the Avogadro number (6.022 × 10^23^ mol^−1^), and h is the Planck constant (6.63 × 10^−34^ Js).

The plot in Figure 6 shows the transition state equation of ln(Cr/T) as a function of 1/T, with and without DPS concentrations. The straight lines were performed with a slope (−$\Delta H_{a}$/R), while the intercept (Ln R/Nh + $\Delta S_{a}$/R) [30], and the values are tabulated in Table 4.

The positive values of $\Delta H_{a}$ indicated the endothermic properties of the carbon steel dissolution process [31]. It is evident that the $\Delta H_{a}$ were shifted in the same way as the $E_{a}$ values, confirming the purposed inhibition mechanism [32]. The large and negative values of $\Delta S_{a}$ indicating that a reduction in disordering takes place on going from reactants to the active complex. This investigation is in harmony with the other authors’ findings [33,34].

#### 3.3.2. Adsorption Isotherm

Adsorption isotherms can be used to investigate the interactions of DPS inhibitor molecules with the carbon steel surface [35]. To identify the adsorption type, various isotherms were considered, including Temkin and Langmuir isotherms. The Langmuir adsorption isotherm model was discovered to be the most suitable model. The linear relationships of C/θ versus C for DPS inhibitor on carbon at different temperatures are shown in Figure 7.

The linear regression coefficients (R^2^) values were almost equal to 1, as shown in Table 5. This indicated that DPS adsorption on the carbon steel surface adhered to the Langmuir adsorption isotherm, which can be expressed as follows [36]:(13)$$\frac{C_{inh}}{\theta }=\frac{1}{K_{ads}}+C_{inh}$$
where $K_{\mathrm{ads}}$ is a constant balance of absorption, C is the concentration of inhibitor (ppm) and the degree of surface coverage determined from weight loss data [37]:(14)$$\Delta G_{ads}^{o}=-RT\left(55.5\times C_{inh}\right)$$
wherever $\Delta G_{\mathrm{ads}}^{o}$ is the Gibbs energy adsorption, and the absolute temperature defined as T in Kelvin. The 55.5 is water concentration in the solution.

Table 5 shows the negative values of $\Delta G_{\mathrm{ads}}^{o}$, this indicates that DPS inhibitor molecules are adsorbed spontaneously on a carbon steel surface to form a stable layer [38]. In general, the $\Delta G_{\mathrm{ads}}^{o}$ values greater than −40 kJ·mol^−1^ indicate that the adsorption could be as chemisorption. In contrast, a negative value from −20 kJ·mol^−1^ or lower shows physisorption [39,40]. In the present study, the obtained values of $\Delta G_{\mathrm{ads}}^{o}$ are located in the range of −33.45 to −38.41 kJ·mol^−1^. Therefore, this indicated that the DPS could protect the carbon steel corrosion by performing mixture adsorption (physical absorption and chemical adsorption). Meantime, at the temperatures of 318 to 338 K, the process was close to chemisorption as described in Figure 8.

The values of enthalpy of adsorption $\left(\Delta H_{\mathrm{ads}}^{o}\right)$ and entropy of adsorption $\left(\Delta S_{\mathrm{ads}}^{o}\right)$ were obtained by plotting $\Delta G_{\mathrm{ads}}^{o}$ against T, as shown in Figure 9. In this study, the values of $\Delta H_{\mathrm{ads}}^{o}$ and $\Delta S_{\mathrm{ads}}^{o}$ was found to be 27.36 kJ·mol^−1^ and −40.68 Jk^−1^·mol^−1^ respectively. Meanwhile, the positive value of $\Delta H_{\mathrm{ads}}^{o}$ indicates that the adsorption phenomenon is endothermic. The negative value of $\Delta S_{\mathrm{ads}}^{o}$ suggests that the adsorption involves an associative mechanism. Additionally, a negative value of S shows that no major changes in the adsorbent’s internal structures occur throughout the adsorption process [41].

### 3.4. Corrosion Inhibitor in HCl Solution Operational Mechanism

Figure 10 shows the proposed corrosion inhibition mechanism by DPS inhibitor. Corrosion inhibitors will generally be ionised in acidic solutions and acquire a positive charge due to the ionisation process. The ionisation reaction of the DSE inhibitor solution occurs where the water molecule (H_2_O) is separated into a hydrogen cation (H^+^) and the hydroxyl anion (OH^−^) [40].

In this study, date seed extract with various functional groups, such as OH, C=O, and OCH_3_, which are nucleophilic, will be protonated or bonded to H^+^ in an acidic aqueous solution (Figure 10a). Cl^−^ and OH^−^ anions tend to move to more electrophilic cations (Fe^+2^). Thus, the steel surface will be negatively charged by the anions CI^−^ and OH^−^ and undergo corrosion reactions and oxide formation. Next, the proton corrosion inhibitor interacts with the negatively charged steel surface (Figure 10b). The positive charge on the active site of the DPS corrosion inhibitor ties to the negative charge on the steel. These interactions cause the inhibition of the corrosion process. Anion Cl^−^ was found adsorbed by coupling with the inhibitor molecule, where it lies between the steel surface (Fe^2+^) and the corrosion inhibitor (H^+^ inhibitor) through the interaction of ionic bonds and Van der Waals forces. Coupling adsorption usually involves interactions between anions and active sites of the corrosion inhibitor [42].

### 3.5. Quantum Chemical Calculation

The quantum parameters of chemicals listed in Table 6 were calculated to understand the mechanism of inhibition offered by lauric, myristic, oleic, phthalic, caprylic, and palmitic acids on the Fe surface. The performance of an inhibitor can be predicted using its E_HOMO_, E_LUMO_, and ΔE values. The tendency of the inhibitor molecule to contribute electrons to the metal surface is correlated with E_HOMO_, while E_LUMO_ is used to explain the ability of inhibitor molecules to obtain electrons from the metal surface [43]. Inhibitors with higher E_HOMO_ values show a high propensity to donate electrons to the metal surface of the unoccupied d-orbital [44]. The E_HOMO_ increasing values facilitate adsorption on the metal surface, thereby improving the effectiveness of inhibition.

On the other hand, the inhibitors with lower E_LUMO_ values show a high capability to accept electrons. In this research, oleic acid has a high E_HOMO_ value, as shown in Table 6, which indicates a greater tendency to give electrons to the Fe surface. In comparison, the lower E_LUMO_ value of phthalic acid exhibits a higher tendency to allow electrons, followed by the Fe surface.

The energy gap (ΔE), which is defined as the difference between E_LUMO_ and E_HOMO_, is another quantum chemical parameter that can be correlated with inhibition efficiency. The ΔE can elucidate the adsorption reactiveness of the inhibitor molecules with the Fe surface. A lower ∆E value of an inhibitor molecule shows that the molecule has higher reactivity and can have better IE compared to the molecules with higher ΔE. Phthalic acid possesses a low ΔE value, which suggests greater reactivity and strong inhibition performance on the Fe surface. The ΔE of oleic acid is lower than lauric acid, as also found by [44]. Other work reports by [45] on lauric, myristic, oleic and caprylic acid show the ΔE of 7.60 eV for lauric and myristic while oleic, caprylic acid have ΔE of 6.16 eV and 7.62 eV, respectively. The deviation between the HOMO-LUMO energy gap value obtained in this work and the other work is due to the difference in the software used and other calculations parameters such as basis sets and exchange-correlation functional. Due to the lower inhibitor electronegativity (χ) values as compared to the work function of Fe, given as 4.82 eV in Fe, the electrons flow from the inhibitor to the Fe surface. The electronegativity value of oleic acid is the lowest amongst other DPS molecules, although the difference in electronegativity value is too small as compared to those of lauric, myristic, and palmitic acids.

Global hardness (η) and softness (σ) are terms used to describe the resistance of atoms to the deformation of their electron cloud [46]. These are important characteristics for calculating molecular stability and reactivity. A large ΔE exists in hard molecules, while the ΔE is low in soft molecules. A hard molecule is less reactive than a soft molecule because it cannot easily give electrons to the acceptor molecule. Thus, a soft-molecule inhibitor is supposed to be more reactive and has more inhibition efficiency than a hard-molecule inhibitor because soft molecules easily provide electrons to the metal surface. In this research, the values of η and σ of lauric, myristic, caprylic, and palmitic acids show small differences, which suggest that the resistance of the species is similar to the disfigurement of its electron structure. As hard molecules are less interactive than soft molecules, the alikeness of the η and σ values of lauric, myristic, caprylic, and palmitic acids suggests that these acids will have the same reactivity towards the Fe surface as inhibitors. This can also be explained by the small differences in their ΔE values. However, the values of η and σ for oleic acid and phthalic acid are not quite as close to those of other inhibitors, which indicates that they would have different reactivity towards the Fe surface. As shown in Table 6, the lowest hardness value for phthalic acid is 1.740 eV, and the lowest softness value is 0.3783 eV^−1^, which is assumed to be the most effective for a corrosion inhibitor.

The IE% through the electrons transferred (ΔN) was higher for oleic acid, indicating a higher tendency to contribute electrons to the Fe surface. All ∆N values are also shown to be lower than 3.6, and according to Lukovits [46], ∆N value of less than 3.6 indicates higher inhibition efficiency on the metal surface due to higher electron-donating power. If ∆N exceeds 0, electrons are transferred from the inhibitor molecules to the Fe surface. On the other hand, if ∆N is less than 0, the opposite process occurs, and electrons are transferred from the Fe surface to the inhibitor molecules [47]. The results show that the inhibitors examined in this study were electron donors and the Fe surface was the electron acceptor.

The electrophilic attack sites representing the regions where the inhibitor molecule and metal surface show that the highest bonding ability is carried by the HOMO orbital. Nucleophilic attack sites are carried by LUMO orbital; they exhibit an antibonding orbital occurring between inhibitor molecules and metal surface to create a bond of feedback that strengthens the interaction between the inhibitor and the surface of Fe.

Table 7 presents the distribution of HOMO electron density over the inhibitor means that the inhibitor molecule is likely to be active in the donation of electrons to the empty Fe orbital, resulting in efficient corrosion inhibition. The distribution of LUMO density over the molecule, on the other hand, confirms that electrons from the occupied Fe orbitals can be effectively accepted, which is a major factor for interactions between the donor and the acceptor to occur. It can be observed that the HOMO and LUMO of lauric, myristic, phthalic, caprylic, and palmitic acid molecules are mainly localised around the carboxyl (COOH) group. However, in the case of oleic acid, the HOMO was mainly localised on the C=C bond, and this finding is consistent with another report [48]. This can be due to the middle of the oleic acid contain a double bond, so the HOMO was saturated around the area. This is supported by a finding from [49], which found that the major contribution to the HOMO comes from the carbon atoms adjacent to the C=C in the oleic acid side.

Therefore, these sites are the most active for the electron donation-acceptance type of interaction and most likely to facilitate adsorption over the Fe surface. Also, the distribution of HOMO-LUMO indicates that all double bonds in simple molecules serve as active HOMO sites, suggesting their tendency to share electronic charge with the atoms of surface metal. In the case of phthalic acid, which has two groups of carboxyl, the distribution of HOMO and LUMO around its benzene ring is observed over the entire molecule.

## 4. Conclusions

The inhibitor properties of DPS for preventing carbon steel corrosion were investigated using experimental and theoretical methods. The corrosion inhibition of carbon steel in an acidic solution by DPS was proven to be more effective than other inhibitors reported in the previous literature. The inhibition efficiency decreased with an increase in both immersion time and temperature, but it increases with an increase in concentrations of DPS up to 1400 ppm. The surface morphology of the carbon steel sample protected with 1400 ppm inhibitor was smoother compared with the sample exposed to the acid solution. This suggests that a protective layer was formed on the carbon steel surface. The EDX analysis of the carbon steel surface shows that the peaks of Cl and O are high in the absence of DPS inhibitor. The activation energy and enthalpy values suggest that DPS performs a dual adsorption, and the properties were endothermic. The adsorption studies indicated that DPS obeyed the Langmuir adsorption model. In addition, the values of Gibbs free energy signified that the DPS inhibition mechanism worked through mixture adsorption on the carbon steel surface. The results attained from a quantum chemical study using DFT methods indicate that the COOH group and C=C bond are the active sites on DPS molecules and are most likely to be responsible for the adsorption on the Fe surface. The findings obtained from the experimental and theoretical measurements in this study are in good agreement.

## Figures and Tables

**Figure 1 molecules-26-03535-f001:**
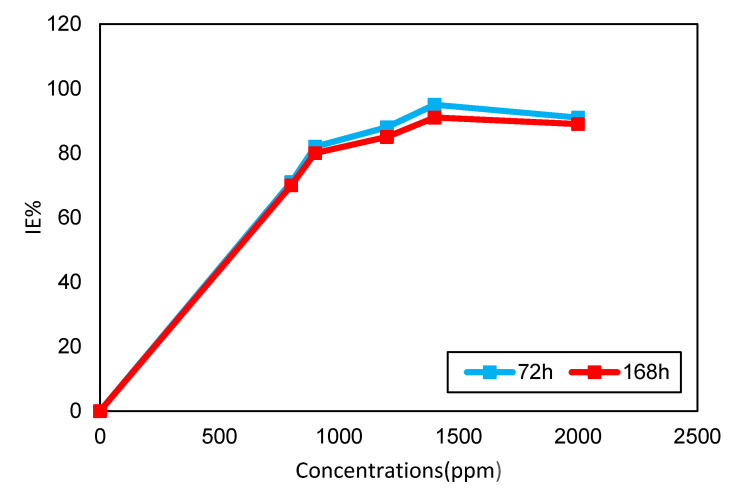
DPS concentrations with IE% at different immersion time (72–168 h).

**Figure 2 molecules-26-03535-f002:**
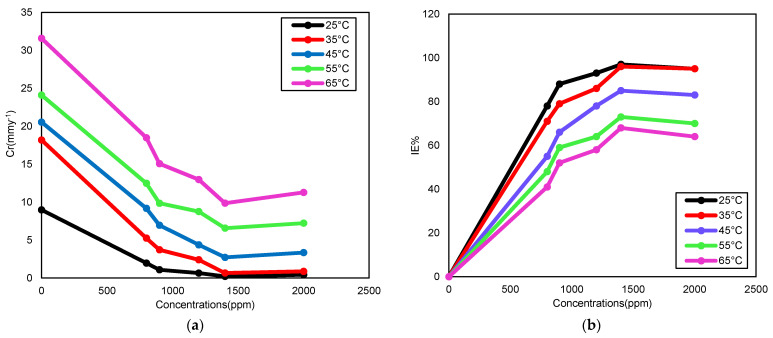
The impact of temperature on (**a**) Cr and (**b**) IE%, after 5 h of immersion in absence and presence DPS.

**Figure 3 molecules-26-03535-f003:**
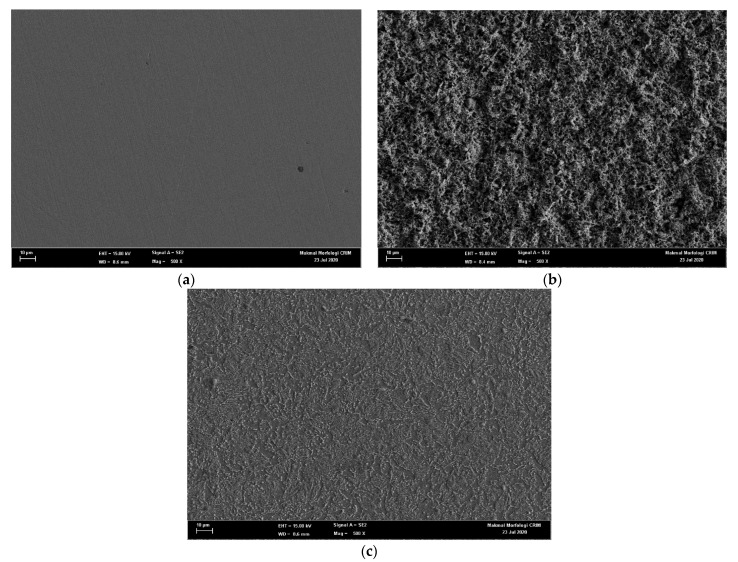
SEM micrographs of carbon steel (**a**) polished sample, (**b**) 0.5 M HCl and (**c**) 1400 ppm of DPS.

**Figure 4 molecules-26-03535-f004:**
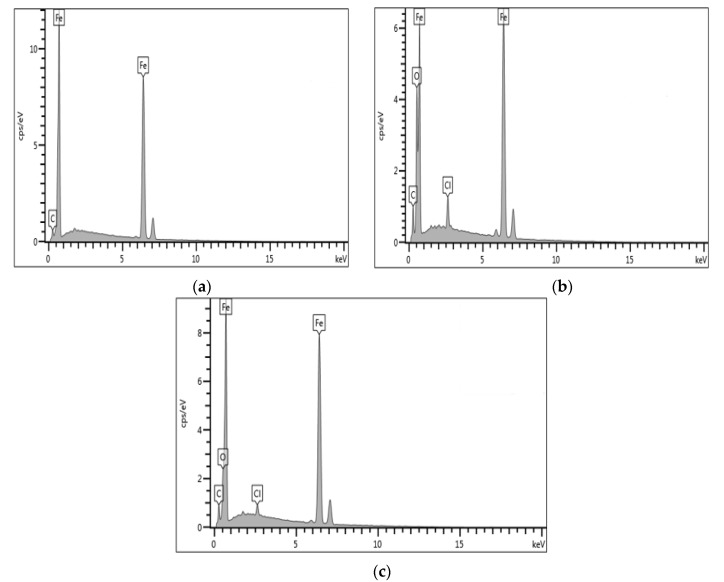
EDX analysis for carbon steel (**a**) polished sample, (**b**) 0.5 M HCl and (**c**) 1400 ppm of DPS.

**Figure 5 molecules-26-03535-f005:**
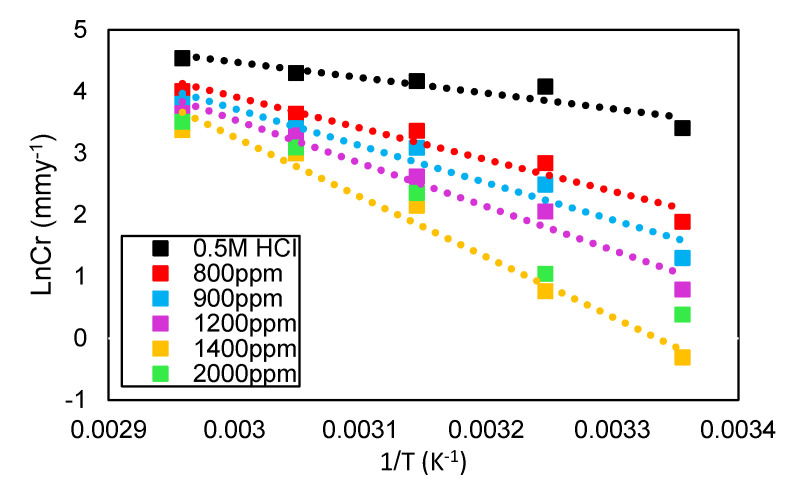
The Arrhenius plot for carbon steel in 0.5 M HCI in the presence and absence of DPS corrosion inhibitor at different temperatures.

**Figure 6 molecules-26-03535-f006:**
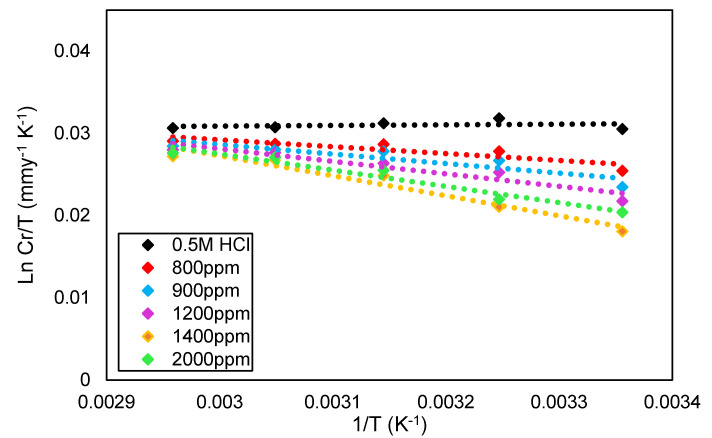
The transition state for carbon steel in 0.5 M HCI in the presence and absence of DPS corrosion inhibitor at different temperatures.

**Figure 7 molecules-26-03535-f007:**
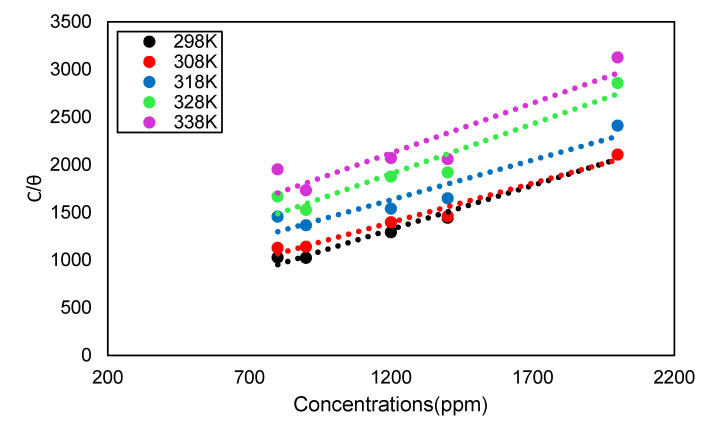
Langmuir adsorption isotherms of DPS on carbon at different temperatures in 0.5 M HCl.

**Figure 8 molecules-26-03535-f008:**
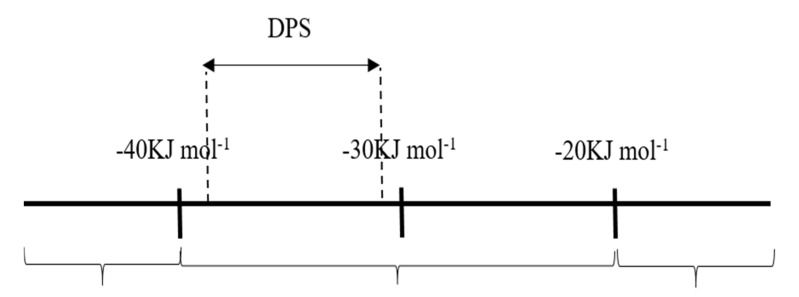
Classification of adsorption type according to $\Delta G_{\mathrm{ads}}^{o}$ values for adsorption of DPS corrosion inhibitor on carbon steel surfaces in 0.5 M HCl solution.

**Figure 9 molecules-26-03535-f009:**
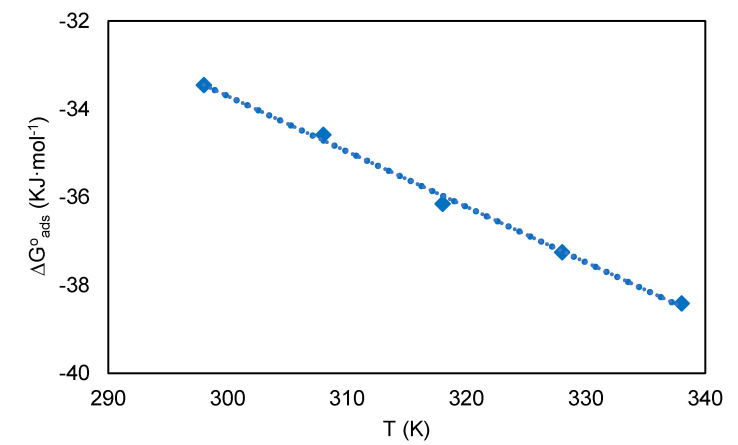
The relationship between $\Delta S_{\;\mathrm{ads}}^{o}$ and temperature (T).

**Figure 10 molecules-26-03535-f010:**
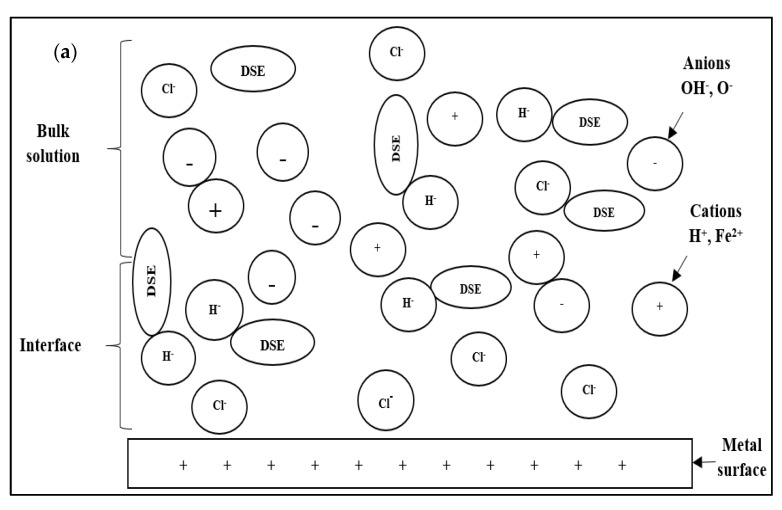
Schematic representation of the corrosion inhibition mechanism in HCI. (**a**) Various ions are dispersed in bulk solution, and Cl^−^ ions move towards the metal interface, (**b**) Inhibitor molecules adsorption and the formation of the protective layer at the optimal inhibitor concentration.

**Table 1 molecules-26-03535-t001:** The acids identified in DPS and their chemical structures.

Symbol	Identified Compounds	Formula	Structure
A	Lauric acid	C_12_H_24_O_2_	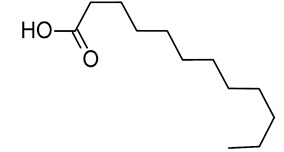
B	Myristic acid	C_14_H_28_O_2_	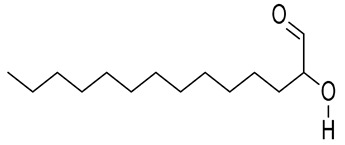
C	Oleic acid	C_18_H_34_O_2_	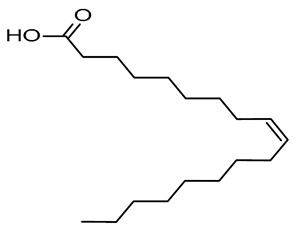
D	Phthalic acid	C_8_H_16_O_2_	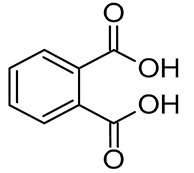
E	Caprylic acid	C_8_H_16_O_2_	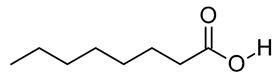
F	Palmitic acid	C_16_H_32_O_2_	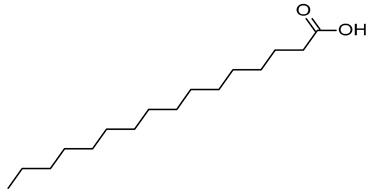

**Table 2 molecules-26-03535-t002:** Weight loss results at different immersion time in the absence and presence of DPS.

Concentrations	Cr (mmy^−1^)	IE%	θ	Cr (mmy^−1^)	IE%	θ
	72 h			168 h		
0.5 MHCl	2.586	-	-	1.304	-	-
800 ppm	0.76	71	0.71	0.391	70	0.70
900 ppm	0.456	82	0.82	0.260	80	0.80
1200 ppm	0.304	88	0.88	0.195	85	0.85
1400 ppm	0.140	95	0.95	0.120	91	0.91
2000 ppm	0.228	91	0.91	0.140	89	0.89

**Table 3 molecules-26-03535-t003:** Weight loss results at various temperatures with and without DPS.

Concentrations			Cr (mmy^−1^)					IE%		
	25 °C	35 °C	45 °C	55 °C	65 °C	25 °C	35 °C	45 °C	55 °C	65 °C
0.5 MHCl	8984	1818	20,542	24,102	31,577	-	-	-	-	-
800 ppm	1972	5258	9183	12,483	18,483	78	71	55	48	41
900 ppm	1095	3724	6964	9860	15,069	88	79	66	59	52
1200 ppm	657	2410	4378	8764	12,973	93	86	78	64	58
1400 ppm	219	661	2730	6573	9864	97	96	85	73	68
2000 ppm	438	876	3357	7230	11,283	95	95	83	70	64

**Table 4 molecules-26-03535-t004:** The activation parameter values of carbon steel in the absence and presence of DPS.

Concentrations	$E_{a}\;\mathrm{kJ}\cdot {\mathrm{mol}}^{-1}$	$\Delta H_{a}\;\mathrm{kJ}\cdot {\mathrm{mol}}^{-1}$	$\Delta S_{a}\;{\mathrm{Jk}}^{-1}\cdot {\mathrm{mol}}^{-1}$
0.5 MHCl	23.62	21.14	−185.49
800 ppm	45.06	42.50	−178.31
900 ppm	52.52	49.88	−175.81
1200 ppm	61.17	58.54	−172.87
1400 ppm	83.43	80.80	−165.12
2000 ppm	72.29	69.67	−169.05

**Table 5 molecules-26-03535-t005:** Parameters calculated using Langmuir adsorption isotherms at various temperatures.

T (K)	R^2^	K_ads_	$\Delta G_{ads}\;\mathrm{kJ}\cdot {\mathrm{mol}}^{-1}$
298	0.98	4.68	−33.45
308	0.973	2.44	−34.58
318	0.952	1.58	−36.15
328	0.921	1.53	−37.25
338	0.90	1.15	−38.41

**Table 6 molecules-26-03535-t006:** Quantum chemical parameters of fatty acids in DPS.

Inhibitor Compounds	Formula	E_HOMO_ (eV)	E_LUMO_ (eV)	ΔE (eV)	IP (eV)	EA (eV)	χ (eV)	η (eV)	σ (eV)^−1^	ΔN
Lauric acid	C_12_H_24_O_2_	−6.255	−0.979	5.276	6.255	0.979	3.617	2.638	0.3791	0.2280
Myristic acid	C_14_H_28_O_2_	−6.258	−0.972	5.286	6.258	0.972	3.615	2.643	0.3784	0.2280
Oleic acid	C_18_H_34_O_2_	−5.539	−0.991	4.548	5.539	0.991	3.265	2.274	0.4398	0.3419
Phthalic acid	C_8_H_6_O_4_	−6.404	−2.924	3.480	6.404	2.924	4.664	1.740	0.5747	0.0448
Caprylic acid	C_8_H_16_O_2_	−6.244	−1.028	5.216	6.244	1.028	3.636	2.608	0.3834	0.2270
Palmitic acid	C_16_H_32_O_2_	−6.260	−0.973	5.287	6.260	0.973	3.617	2.644	0.3783	0.2276

**Table 7 molecules-26-03535-t007:** DFT calculations of optimised structure and HOMO and LUMO structures for DPS fatty acids compounds (atom legend: white = H, light grey = C, and red = O).

Acid Name	Optimised	HOMO	LUMO
Lauric acid	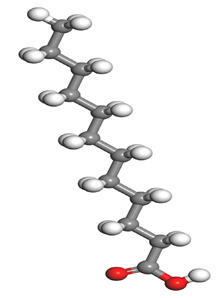	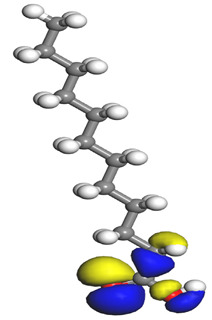	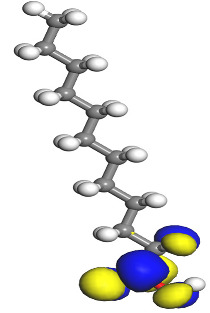
Myristic acid	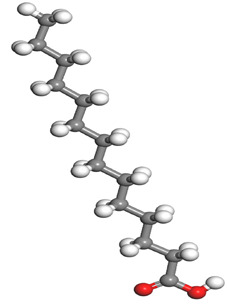	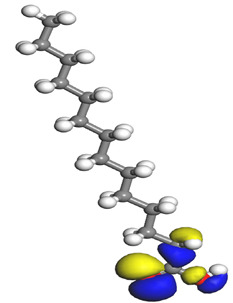	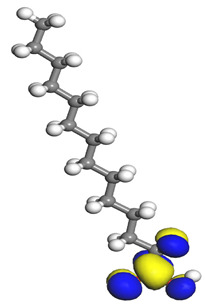
Oleic acid	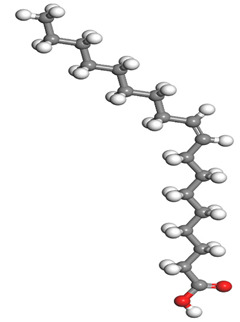	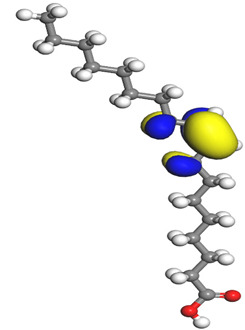	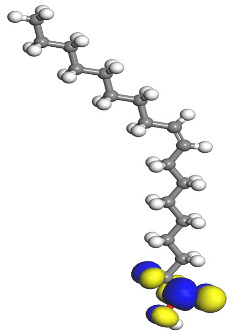
Phthalic acid	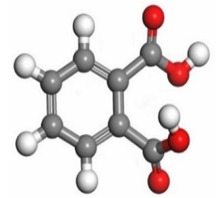	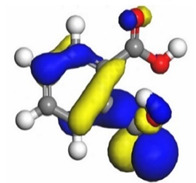	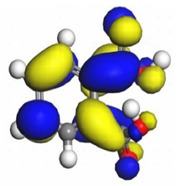
Caprylic acid	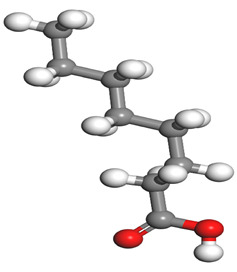	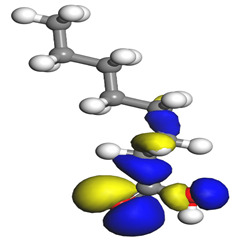	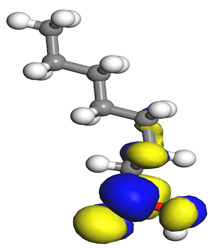
Palmitic acid	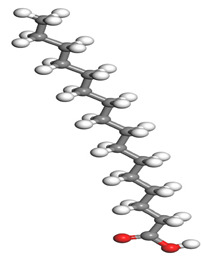	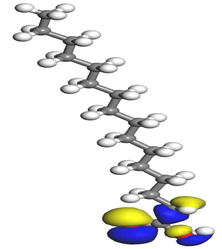	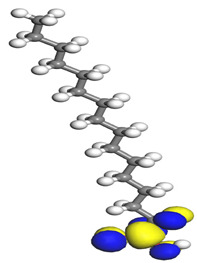

## Data Availability

The data presented in this study are available in this article.

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
