# Peer review of "Experimental and Theoretical Studies on Extract of Date Palm Seed as a Green Anti-Corrosion Agent in Hydrochloric Acid Solution"

_molecules, 2021, doi:10.3390/molecules26123535_

Round 1
Reviewer 1 Report
The Authors try to present DPS as a good choice when trying to stop the corrosion process. In that case they have done some experiments supported by routine calculations. The study needs to be improved, my comments can be found below.
Line 60, why published with capital “P”?
Line 64, isn’t it a strange coincidence that efficiency for those three different materials was exactly 89%?
Further, how is this “efficiency” measured? In other words, what does it mean “89% efficiency”?
Introduction, I surely appreciate the application of “green materials” ad corrosion inhibitors. But to be honest and to give a better perspective for the Readers the disadvantages of such solution must be presented as well.
Table 1, myristic acid structure is wrong.
Table 1, Instead of “name” it should be “formula” and instead of “formula” it should be “structure”.
Table 1, the structures of oleic, caprylic and palmitic acid are wrong.
Table 1, phthalic acid is not a fatty acid, as the Table captions says.
Table 1, could you please provide some information about the typical content (in % or promiles) of those acids in DPS?
Line 116, “2” after “cm” should be removed.
In the introduction, a typical composition of DPS should be presented. I understand the in that case the Authors believe that the acids are the active substance but DPS is composed of other compounds (sugars, proteins, etc.) as well, co this must be introduced.
Lines 151-156, those are well known information and don’t have to be presented here
Line 165, have you used dispersion correction? This is very important.
Line 197, how can you be sure about this?
Figure 5, the linear equation based on solely three experimental points can be very inaccurate. Those are experimental values, so the line equation should be based on at least five points (five temperatures).
Discussion, the Authors discuss the types of adsorption, but the possibility of chemical absorption is this case is highly doubtful. What type of chemical interactions could possible be between the metal and acid molecules? Metal is not a typical adsorbent because of its inability to form long lasting interactions with most of organics.
Figure 8, once again a line made of three points, please increase to at least five.
Calculations. While I strongly support the molecular modelling methods, in this particular case I found it strange that the Authors have use them in such way. Instead of performing molecular dynamics simulations, which would be interesting and could possible prove the hypothesis stated in this paper, the Authors focus on calculations of basic properties of simple organic acids. Those results are useless, in this case, without the information about the CONTENT of those compounds in DPS, or, at least, their relative content. Without such information the Authors can model every single acid they think of and the conclusions would be the same.
However, if the Authors have done the calculations they should do some literature research and COMPARE the obtained results for those very simple and popular compounds with the results of the same calculations on the same compound but maybe done using different software or DFT functionals. I am sure someone before has calculated HOMO/LUMO and those other basic parameters using B3LYP in Gaussian. After all the studied molecules are very important and common. Please compare.
Author Response
|
Point 1: Line 60, why published with capital “P”? Response 1: (Revised as requested). The P of (published) in line 60 has been changed from capital to small letter.
|
|
Point 2: Line 64, isn’t it a strange coincidence that efficiency for those three different materials was exactly 89%? Response 2: (Revised as requested). After checking the efficiency values obtained from previous authors, it was found that the IE for Papaver somniferum in ref (7) is 83% instead of 89%. |
|
Point 3: Further, how is this “efficiency” measured? In other words, what does it mean “89% efficiency”? Response 3: (Revised as requested). This efficiency has been measured by using corrosion measurements such as weight loss and impedance. The efficiency of 89% that has been mentioned in the previous work means the ability of the inhibitor to protect the metal from corrosion. |
|
Point 4: Table 1, myristic acid structure is wrong. Response 4: (Revised as requested). The myristic acid structure in Table 1 has been corrected.
|
|
Point 5: Table 1, Instead of “name” it should be “formula” and instead of “formula” it should be “structure”. Response 5: (Revised as requested). The “name” in Table1 has been changed to “formula” and “formula” replaced to “structure”.
|
|
Point 6: Table 1, the structures of oleic, caprylic and palmitic acid are wrong. Response 6: (Revised as requested). The oleic, caprylic, and palmitic acid structures in Table 1 has been corrected. |
|
Point 7: Table 1, phthalic acid is not a fatty acid, as the Table captions says. Response 7: (Revised as requested). |
|
Point 8: Table 1, could you please provide some information about the typical content (in % or promiles) of those acids in DPS? Response 8: (Revised as requested). Information about the importance of these acids has been explained as follows: The importance of this class of acids is due to their molecular features. Believably, the functional group in fatty acids enhances the surface chemistry toward interaction with metal-vacant orbitals and plays a crucial role in improving the adsorption characteristics. Furthermore, the active group of fatty acid molecules may interact with the metal casing to protect the metal from corrosion. Hence, they contain absorbable carboxyl functional groups for attachment to metal surfaces. (This information has been added to the manuscript). |
|
Point 9: Line 116, “2” after “cm” should be removed. Response 9: (Revised as requested). The “2” in Line 116 has been removed. |
|
Point 10: In the introduction, a typical composition of DPS should be presented. I understand the in that case the Authors believe that the acids are the active substance, but DPS is composed of other compounds (sugars, proteins, etc.) as well, co this must be introduced. Response 10: The compounds of date palm fruit are sugars, proteins, etc. While, the main compounds of date palm seed were fatty acid, which identified by GCMS and the result listed in Table 1 in the manuscript. Some previous work has obtained similar GCMS results for the date palm seed compounds [1]–[3]. |
|
Point 11: Lines 151-156, those are well known information and don’t have to be presented here. Response 11: (Revised as requested). The information in line 151-156 has been removed. |
|
Point 12: Line 165, have you used dispersion correction? This is very important. Response 12: We did not use the dispersion correction in this research study. |
|
Point 13: Line 197, how can you be sure about this? Response 13: The explanation in line 197 has been deeply supported by different previous work [4][5]. Therefore, these works have been added as references in line 196-197 in the manuscript. |
|
Point 14: Figure 5, the linear equation based on solely three experimental points can be very inaccurate. Those are experimental values, so the line equation should be based on at least five points (five temperatures). Response 14: (Revised as requested). Two different temperature (45 and 65oC) has been added so that the manuscript was conducted on five different temperatures (as your request). Therefore Figure 2, 5, 6, 7 and 8 have been updated based on added studies; likewise, for the data in Tables 3, 4 and 5. |
|
Point 15: Discussion, the Authors discuss the types of adsorption, but the possibility of chemical absorption is this case is highly doubtful. What type of chemical interactions could possibly be between the metal and acid molecules? Metal is not a typical adsorbent because of its inability to form long lasting interactions with most of organics. Response 15: In corrosion inhibitor studies, the inhibitory process involves the interaction of transport of inhibitor molecules to the metal side, followed by the interaction of the molecule with the metal surface. When a corrosion inhibitor is added to the corrosion medium, the electric structure will change where the inhibitor molecule will take place and replace the base of the water molecule that has been adsorbed on the metal surface as in the equation below:
Where “sol” represents the aqueous solution, “ads” are the adsorption phase, and n is the number of water molecules that need to be absorbed from the metal surface to replace the inhibitor molecule. This statement is also supported by [6], who found that the replacement of water molecules by the adsorption reaction of inhibitor molecules on metal surfaces has reduced the process of metal dissolution. |
|
Point 16: Figure 8, once again, a line made of three points, please increase to at least five. Response 16: (Revised as requested). Two different temperature (45 and 65oC) has been added so that the manuscript was conducted on five different temperatures (as your request). Therefore Figure 2, 5, 6, 7 and 8 have been updated based on added studies; likewise, for the data in Tables 3, 4 and 5. |
|
Point 17: However, if the Authors have done the calculations they should do some literature research and COMPARE the obtained results for those very simple and popular compounds with the results of the same calculations on the same compound but maybe done using different software or DFT functional. I am sure someone before has calculated HOMO/LUMO and those other basic parameters using B3LYP in Gaussian. After all the studied molecules are very important and common. Please compare. Response 16: The literature research has been done to compare with our results. However, not much literature for all the acid was found to make the comparison.
|
*Notice: The Arrhenius plot has been added to the manuscript to clarify the Thermodynamic and Kinetic studies. The plot was added under the heading of "Figure 5. The Arrhenius plot for carbon steel in 0.5 M HCI in the presence and absence of DPS corrosion inhibitor at different temperatures."
References.
[1] I. A. Nehdi, H. M. Sbihi, C. P. Tan, U. Rashid, and S. I. Al-Resayes, “Chemical Composition of Date Palm (Phoenix dactylifera L.) Seed Oil from Six Saudi Arabian Cultivars,” J. Food Sci., vol. 83, no. 3, pp. 624–630, 2018, doi: 10.1111/1750-3841.14033.
[2] S. Besbes, C. Blecker, C. Deroanne, N. E. Drira, and H. Attia, “Date seeds: Chemical composition and characteristic profiles of the lipid fraction,” Food Chem., vol. 84, no. 4, pp. 577–584, 2004, doi: 10.1016/S0308-8146(03)00281-4.
[3] H. Ogungbenle, “Chemical and Fatty Acid Compositions of Date Palm Fruit (Phoenix dactylifera L) Flour,” Bangladesh J. Sci. Ind. Res., vol. 46, no. 2, pp. 255–258, 1970, doi: 10.3329/bjsir.v46i2.8194.
[4] M. G. Tsoeunyane, M. E. Makhatha, and O. A. Arotiba, “Corrosion Inhibition of Mild Steel by Poly(butylene succinate)-L-histidine Extended with 1,6-diisocynatohexane Polymer Composite in 1 M HCl,” Int. J. Corros., vol. 2019, 2019, doi: 10.1155/2019/7406409.
[5] M. A. Quaraishi, D. Jamal, and M. Tariq Saeed, “Fatty acid derivatives as corrosion inhibitors for mild steel and oil‐well tubular steel in 15% boiling hydrochloric acid,” J. Am. Oil Chem. Soc., vol. 77, no. 3, pp. 265–268, 2000, doi: 10.1007/s11746-000-0043-3.
[6] M. Benabdellah et al., “Inhibitive action of some bipyrazolic compounds on the corrosion of steel in 1 M HCl. Part I: Electrochemical study,” Mater. Chem. Phys., vol. 105, no. 2–3, pp. 373–379, 2007, doi: 10.1016/j.matchemphys.2007.05.001.
Reviewer 2 Report
The potential to use date palm seed as a anticorrosive coating is interesting. The overall methods used by the authors are appropriate. However, there are a number of issues that need to be addressed in a revised manuscript. These points are listed below.
- Line 116: The '2' after cm should be superscripted.
- Line: 138: The value of 87600 needs to be discussed. (24 h x 356 d = 8760).
- Line 150-159: Additional detail about the SEM conditions is needed as well as the details of the EDX analysis.
- Line 173: The units of the work function for iron are needed. I assume eV.
- Figures 1 and 2: On the x-axis, the units are usually given along with Concentration and not with every division.
- Figure 3: The scale for each micrograph is barely readable, even the manuscript is enlarged by 300%. The authors should state something about the total size of each image; 50 x 50 μm for example.
- Line 263: The value for Avogadro's number is incorrect, instead the value of R is given.
- Table 4: The units for entropy are incorrect.
- Figure 6: The x-axis has no units. They are needed.
- Figure 7: -30 kJ mol-1 (The '1' is missing)
- Figure 8: I believe the x-axis is 1/T and not "T"
- Table 6: These calculated values need to have units where appropriate.
Author Response
|
Response to Reviewer 2 Comments Point 1: Line 116: The '2' after cm should be superscripted. Response 1: (Revised as requested). |
|
Point 2: Line: 138: The value of 87600 needs to be discussed. (24 h x 356 d = 8760). Response 2: (Revised as requested). |
|
Point 3: Line 150-159: Additional detail about the SEM conditions is needed as well as the details of the EDX analysis. Response 3: (Revised as requested). The details about the SEM and EDX conditions has been added. |
|
Point 4: Line 173: The units of the work function for iron are needed. Response 4: The units of the work function for iron has been added. |
|
Point 5: Figures 1 and 2: On the x-axis, the units are usually given along with Concentration and not with every division. Response 5: The units have been removed from the division. |
|
Point 6: Figure 3: The scale for each micrograph is barely readable, even the manuscript is enlarged by 300%. The authors should state something about the total size of each image; 50 x 50 μm for example. Response 6: (Revised as requested). The images have been replaced with clear ones (section 3.2); in addition to that, the size of each image has been included in the SEM Methodology (section 2.4). |
|
Point 7: Line 263: The value for Avogadro's number is incorrect, instead the value of R is given. Response 7: (Revised as requested). The Avogadro's number has been replaced with correct value (6.022×1023 mol-1). |
|
Point 8: Table 4: The units for entropy are incorrect. Response 8: The entropy values have been updated based on the latest temperatures test (five temperature). |
|
Point 9: Figure 6: The x-axis has no units. They are needed. Response 9: The x-axis unit (ppm) has been added. |
|
Point 10: Figure 7: -30 kJ mol-1 (The '1' is missing) Response 10: The 1 in value has been added. |
|
Point 11: Figure 8: I believe the x-axis is 1/T and not "T" Response 11: There are different plots to calculate and , 1. By plotting against T Basic Thermodynamic equation [1], [2]. 2. By plotting against 1/T (Gibbs- Helmholtz Equation) [3]. Therefore this manuscript used Basic Thermodynamic equation. |
|
Point 12: Table 6: These calculated values need to have units where appropriate. Response 12: The unit for the calculated values have been added.
*Notice: The Arrhenius plot has been added to the manuscript to clarify the Thermodynamic and Kinetic studies. The plot was added under the heading of "Figure 5. The Arrhenius plot for carbon steel in 0.5 M HCI in the presence and absence of DPS corrosion inhibitor at different temperatures"
References.
[1] N. Labjar, F. Bentiss, M. Lebrini, C. Jama, and S. El Hajjaji, “Study of temperature effect on the corrosion inhibition of C38 carbon steel using amino-tris(methylenephosphonic) acid in hydrochloric acid solution,” Int. J. Corros., vol. 2011, no. October, 2011, doi: 10.1155/2011/548528. [2] M. P. Desimone, G. Gordillo, and S. N. Simison, “The effect of temperature and concentration on the corrosion inhibition mechanism of an amphiphilic amido-amine in CO2 saturated solution,” Corros. Sci., vol. 53, no. 12, pp. 4033–4043, 2011, doi: 10.1016/j.corsci.2011.08.009. [3] P. B. Matad, P. B. Mokshanatha, N. Hebbar, V. T. Venkatesha, and H. C. Tandon, “Ketosulfone drug as a green corrosion inhibitor for mild steel in acidic medium,” Ind. Eng. Chem. Res., vol. 53, no. 20, pp. 8436–8444, 2014, doi: 10.1080/17518250903170868. |
Reviewer 3 Report
Referee Answer:
In this paper, the author tried to estimate the redox potential of a palm seed extract and its potential using as an anti-corrosive agent. In addition, to be more insight into the process, the authors used conceptual density functional theory. Even though the manuscript has a suitable science-sound and science appeal, some details must address before publishing.
- The abstract can be improved. I am pretty sure the results found in this manuscript are promising; thus, the abstract has to be more convincing; for example, from molecular modelling, the author found some evidence about the mechanism of action of this extract. However, this is not highlighted in the abstract. In addition, some “typos” must be corrected; the abstract does not have to do abbreviation.
- In the introduction, the statement “Fatty acids are attractive candidates as corrosion inhibitors because they contain adsorbable carboxyl functional groups for attachment to metal surfaces” the world attachment does not match with the idea; moreover, this statement must be supported with bibliography. In addition, the main goal of this work is not clear. What is the principal idea, theoretical or experimental ones? If the corrosion-inhibition is already done for this extract, what this work goes for? What is the sense to use computational chemistry joint to inhibitor study? Please, clarified these point.
- The authors calculated the IP and EA using the Koopmans' theorem. However, from philosophy of DFT, this theorem might not be applying to large molecules such fatty acids. Please, add any explanation why do you use this theorem instead IP and EA vertical approximations?
- The results have some merits; however, the use of only three point diminishing the accurate of the Gibbs models. We strongly recommend added, at least, two more points.
- Table 5 shows the adsorption equilibrium decreased with temperature. Please, add the plausible chemical explanation for this phenomenon.
- In table 6, the authors report a dual molecular description, D However, the authors did not clarify the electron fluxes direction. In addition, the fatty acids are the responsibility of the inhibition, did this result related to the redox potential of fatty acid? Please, clarified this.
- In table 4, the activation entropy increases with concentration. Why the authors explain this behavior.
- In table 7, the entire homo orbitals rest in the carboxylic acid moiety. However, the HOMO of oleic acid rests in the middle o the proteins. Please, supply some explanation for this behavior.
- Finally, the conclusion is a little vague. Please, write a most convincing conclusion.
I am confident the author can address this suggestion and send back the manuscript.
Thanks very much!
Author Response
Response to Reviewer 3 Comments
|
Point 1: The abstract can be improved. I am pretty sure the results found in this manuscript are promising; thus, the abstract has to be more convincing; for example, from molecular modelling, the author found some evidence about the mechanism of action of this extract. However, this is not highlighted in the abstract. In addition, some “typos” must be corrected; the abstract does not have to do abbreviation. Response 1: (Revised as requested).
|
|
Point 2: In the introduction, the statement “Fatty acids are attractive candidates as corrosion inhibitors because they contain adsorbable carboxyl functional groups for attachment to metal surfaces” the world attachment does not match with the idea; moreover, this statement must be supported with bibliography. In addition, the main goal of this work is not clear. What is the principal idea, theoretical or experimental ones? If the corrosion-inhibition is already done for this extract, what this work goes for? What is the sense to use computational chemistry joint to inhibitor study? Please, clarified these point. Response 2: the main goals of this work are as follows: 1. To determine and evaluate the corrosion inhibitor efficiency of date palm seed extract in 0.5M HCl medium using weight loss technique. 2. To determine the effect of temperature and immersion time on the corrosion rate and percentage of inhibition efficiency. 3. To Identify the mechanism of date palm seed corrosion inhibition in 0.5M HCl. 4. To conduct quantum chemical calculations on date palm seed inhibitor using the DFT method to calculate several quantum chemical parameters to evaluate the inhibitor performance theoretical. (All the mentioned goals have been achieved and proven in the manuscript).
What is the sense to use computational chemistry joint to inhibitor study? Please, clarified these point. Using experimental methods to investigate the relationship between corrosion inhibitor and inhibition efficiency is not enough and costly, time-consuming. It is economical and fast to apply computational techniques as predictive techniques such as quantum chemical calculations and density functional theory (DFT). Theoretical studies methods can be complemented the experimental investigations and be practical tools to propose the best compound in corrosion inhibitors. Therefore, Quantum chemical was conducted to study the dependence between the DPS inhibiting properties and their structure using DFT methods. |
|
Point 3: The authors calculated the IP and EA using the Koopmans' theorem. However, from philosophy of DFT, this theorem might not be applying to large molecules such fatty acids. Please, add any explanation why do you use this theorem instead IP and EA vertical approximations? Response 3: DFT develops the molecular orbitals on a valence basis set. The IP and EA using Koopmans' theorem also found in other work related to a corrosion inhibitor. [1], [2]. The Koopman’s theorem has been used to roughly estimate the IP and EA of the fatty acid [3]. |
|
Point 4: The results have some merits; however, the use of only three point diminishing the accurate of the Gibbs models. We strongly recommend added, at least, two more points. Response 4: (Revised as requested). Two different temperature (45 and 65oC) has been added so that the manuscript was conducted on five different temperatures (as your request). Therefore, Figure 2, 5, 6, 7 and 8 have been updated based on added studies; likewise, for the data in Tables 3, 4 and 5. |
|
Point 5: Table 5 shows the adsorption equilibrium decreased with temperature. Please, add the plausible chemical explanation for this phenomenon. Response 5: The negative values of indicate that DPS inhibitor molecules are adsorbed spontaneously on a carbon steel surface to form a stable layer. The has been changed based on the latest temperature study. The increase in temperature makes the adsorption close to being chemical adsorption [4]. (This explanation has been added to the manuscript from line 308 to 301). |
|
Point 6: In table 6, the authors report a dual molecular description, D However, the authors did not clarify the electron fluxes direction. In addition, the fatty acids are the responsibility of the inhibition, did this result related to the redox potential of fatty acid? Please, clarified this. Response 6: Thank you for your question. However, the increases and reduction in Fatty Acids are not our main goal in this research study. Our main goal was clearly discussed in point 2. |
|
Point 7: In table 4, the activation entropy increases with concentration. Why the authors explain this behaviour. Response 7: The activation entropy increased with concentration, indicating that a reduction in disordering takes place on going from reactants to the active complex (this will retard the discharge of hydrogen ions at the metal surface, causing the system to pass from a random arrangement). This reduction in disordering increase with an increase in the concentrations. This investigation is in harmony with the other authors' findings [5], [6]. |
|
Point 8: In table 7, the entire homo orbitals rest in the carboxylic acid moiety. However, the HOMO of oleic acid rests in the middle of the proteins. Please, supply some explanation for this behaviour. Response 8: (Revised as requested). This is because the middle of the oleic acid contains a double bond, so the HOMO was saturated around the area. We have added the explanation in the discussion. |
|
Point 9: Finally, the conclusion is a little vague. Please, write a most convincing conclusion. Response 9: (Revised as requested). |
*Notice: The Arrhenius plot has been added to the manuscript to clarify the Thermodynamic and Kinetic studies. The plot was added under the heading of "Figure 5. The Arrhenius plot for carbon steel in 0.5 M HCI in the presence and absence of DPS corrosion inhibitor at different temperatures."
References.
[1] S. Salaji and N. H. Jayadas, “Experimental and Molecular Level Analysis of the Tribological and Oxidative Properties of Chaulmoogra Oil,” Adv. Tribol., vol. 2020, 2020, doi: 10.1155/2020/8821316.
[2] D. Valencia, I. García-Cruz, V. H. Uc, L. F. Ramírez-Verduzco, M. A. Amezcua-Allieri, and J. Aburto, “Unravelling the chemical reactions of fatty acids and triacylglycerides under hydrodeoxygenation conditions based on a comprehensive thermodynamic analysis,” Biomass and Bioenergy, vol. 112, no. August 2017, pp. 37–44, 2018, doi: 10.1016/j.biombioe.2018.02.014.
[3] K. Wang et al., “Relationship between the electrical characteristics of molecules and fast streamers in ester insulation oil,” Int. J. Mol. Sci., vol. 21, no. 3, 2020, doi: 10.3390/ijms21030974.
[4] M. M. Fares, A. K. Maayta, and M. M. Al-Qudah, “Pectin as promising green corrosion inhibitor of aluminum in hydrochloric acid solution,” Corros. Sci., vol. 60, pp. 112–117, 2012, doi: 10.1016/j.corsci.2012.04.002.
[5] N. Labjar, F. Bentiss, M. Lebrini, C. Jama, and S. El Hajjaji, “Study of temperature effect on the corrosion inhibition of C38 carbon steel using amino-tris(methylenephosphonic) acid in hydrochloric acid solution,” Int. J. Corros., vol. 2011, pp. 1–9, 2011, doi: 10.1155/2011/548528.
[6] E. I. Ating, S. A. Umoren, I. I. Udousoro, E. E. Ebenso, and A. P. Udoh, “Leaves extract of ananas sativum as green corrosion inhibitor for aluminium in hydrochloric acid solutions,” Green Chem. Lett. Rev., vol. 3, no. 2, pp. 61–68, 2010, doi: 10.1080/17518250903505253.
Round 2
Reviewer 1 Report
Though the Authors have revised their manuscript, some of the structures (B, E, F) in Table 1 are still wrong.
Author Response
|
Reviewer 1: Point 1: Though the Authors have revised their manuscript, some of the structures (B, E, F) in Table 1 are still wrong. Response 1: The structures (B, E, F) in Table 1 have been corrected and highlighted with green colour.
|
Reviewer 2 Report
The authors have addressed many of the previous concerns in this revised manuscript. The manuscript will be to be proofread for English and grammar. There are still two concerns as listed below.
- The units of entropy are J K-1 or for molar entropy, J K-1 mol-1. The needs to be corrected in the manuscript.
- The formatting of the references should be in accordance with the Guidelines for Authors for this journal.
Author Response
|
Reviewer 2: Point 1: The units of entropy are J K-1 or for molar entropy, J K-1 mol-1. The needs to be corrected in the manuscript. Response 1: The units of entropy in Table 2 and line 320 have been updated from kJ.mol-1 to Jk-1.mol-1 and highlighted with green colour. Point 2: The formatting of the references should be in accordance with the Guidelines for Authors for this journal. Response 2: The references formatting has been checked and the reference (7, 8, 9, 11, 22, 27, and 49) have been updated based on Molecules Journal formatting and highlighted with green colour.
* The manuscript's language already proofread with professional proofreading by the UK, and the certificate has been uploaded to the system. |
Reviewer 3 Report
The authors have addressed almost all suggestions; however, point 6 is not clarified; I strongly recommend address this point. Even there is not one of the goals of the manuscript, they used fatty acid as the main compound, represent their frontier orbitals and their relationship with the redox process. On the other hand, fatty acids are a cornerstone of redox potential. Please, address this point.
Author Response
|
Reviewer 3: Point 1: The authors have addressed almost all suggestions; however, point 6 is not clarified; I strongly recommend address this point. Even there is not one of the goals of the manuscript, they used fatty acid as the main compound, represent their frontier orbitals and their relationship with the redox process. On the other hand, fatty acids are a cornerstone of redox potential. Please, address this point. Response 1: The Fatty acid compounds of DPS displayed corrosion inhibition characteristics on the redox electrochemical process basically through adsorption. Its presence in the acid media stifled the oxygen reduction, hydrogen evolution and oxidation reaction mechanism responsible for corrosion Adsorption of DPS molecules onto the carbon steel surface blocked the active sites where the dissolution and release of metal cations into the solution occurs as a result of the chloride anions. Corrosion inhibitors will generally be ionised in acidic solutions and acquire a positive charge due to the ionisation process. The ionisation reaction of the DSE inhibitor solution occurs where the water molecule (H2O) is separated into a hydrogen cation (H+) and the hydroxyl anion (OH¯) (Loto 2017). In this study, date seed extract with various functional groups, such as OH, C=O, and OCH3, which are nucleophilic, will be protonated or bonded to H+ in an acidic aqueous solution. Cl¯ and OH¯ anions tend to move to more electrophilic cations (Fe+2). Thus, the steel surface will be negatively charged by the anions CI¯ and OH¯ and undergo corrosion reactions and oxide formation. Next, the proton corrosion inhibitor interacts with the negatively charged steel surface. The positive charge on the active site of the DPS corrosion inhibitor ties to the negative charge on the steel. These interactions cause the inhibition of the corrosion process. Anion Cl¯ was found adsorbed by coupling with the inhibitor molecule, where it lies between the steel surface (Fe2+) and the corrosion inhibitor (H+ inhibitor) through the interaction of ionic bonds.
*This already explained in sub-section 3.4.
|